# SLAM Family Receptors in B Cell Chronic Lymphoproliferative Disorders

**DOI:** 10.3390/ijms25074014

**Published:** 2024-04-04

**Authors:** Dominik Kľoc, Slavomír Kurhajec, Mykhailo Huniadi, Ján Sýkora, Tomáš Guman, Marek Šarišský

**Affiliations:** 1Department of Pharmacology, Faculty of Medicine, Pavol Jozef Šafárik University in Košice, Trieda SNP 1, 04011 Košice, Slovakia; dominik.kloc@student.upjs.sk (D.K.); guniadi97@gmail.com (M.H.); 2Department of Pharmaceutical Technology, Pharmacognosy, and Botany, University of Veterinary Medicine and Pharmacy, Komenského 73, 04181 Košice, Slovakia; slavomir.kurhajec@uvlf.sk; 3Department of Haematology and Oncohaematology, Faculty of Medicine, Pavol Jozef Šafárik University in Košice and Louis Pasteur University Hospital Košice, Trieda SNP 1, 04011 Košice, Slovakia; jan.sykora@unlp.sk (J.S.); tomas.guman@upjs.sk (T.G.)

**Keywords:** SLAM family receptors, SLAMF, SAP, EAT-2, B cells, B cell chronic lymphoproliferative disorders, B-CLPD, leukemia, lymphoma, flow cytometry

## Abstract

The signaling lymphocytic activation molecule (SLAM) receptor family (SLAMF) consists of nine glycoproteins that belong to the CD2 superfamily of immunoglobulin (Ig) domain-containing molecules. SLAMF receptors modulate the differentiation and activation of a wide range of immune cells. Individual SLAMF receptors are expressed on the surface of hematopoietic stem cells, hematopoietic progenitor cells, B cells, T cells, NK cells, NKT cells, monocytes, macrophages, dendritic cells, neutrophils, and platelets. The expression of SLAMF receptors was studied during normal B cell maturation. Several SLAMF receptors were also detected in cancer cell lines of B-lymphoid origin and in pathological B cells from patients with B cell chronic lymphoproliferative disorders (B-CLPD), the most frequent hematological malignancies in adults. This review summarizes current knowledge on the expression of SLAMF receptors and their adaptor proteins SAP and EAT-2 in B-CLPD. Several SLAMF receptors could be regarded as potential diagnostic and differential diagnostic markers, prognostic factors, and targets for the development of novel drugs for patients with B-CLPD.

## 1. Introduction

The SLAM family receptors belong to the CD2 superfamily of immunoglobulin (Ig) domain-containing molecules and are known to be widely expressed by hematopoietic cells. The SLAM family includes nine receptors: SLAMF1 (CD150, SLAM, or IPO-3), SLAMF2 (CD48, BCM1, Blast-1, or HuLy-m3), SLAMF3 (CD229 or Ly-9), SLAMF4 (CD244, 2B4, or NAIL), SLAMF5 (CD84, Ly9b, or GR6), SLAMF6 (CD352, Ly108, NTB-A, SF2000, or TCOM), SLAMF7 (CD319, CS1, CRACC, or 19A24), SLAMF8 (CD353, BLAME, or SBBI42), and SLAMF9 (CD84-H1, SF2001, or CD2F10) [1]. SLAMF2/CD48, SLAMF8/CD353, and SLAMF9/CD84-H1 can be designated as atypical SLAMF receptors due to differences in their cytoplasmic domains in comparison with the other SLAMF receptors.

The SLAMF receptors are type I glycoproteins. The extracellular ectodomain of all SLAMF receptors except SLAMF3/CD229 typically consists of two Ig-like domains: one amino terminal variable (V)-like lacking disulfide bonds followed by a truncated Ig constant 2 (C2)-like domain with two intradomain disulfide bonds (Figure 1) [2]. However, SLAMF3/CD229 possesses four Ig-like domains (two tandem repeats of V-Ig/C2-Ig sets) [3,4,5]. The intracellular tails of the typical SLAMF receptors contain multiple immunoreceptor tyrosine-based switch motifs (ITSMs) with a characteristic homology sequence TxYxxV/I/L/T (where x is any amino acid) across ITSMs [6,7]. These ITSMs bind SLAM-associated protein (SAP) and Ewing’s sarcoma transcript-2 (EAT-2) [8]. As for the atypical SLAMF receptors, SLAMF2/CD48 is embedded in the cell membrane by means of a glycosylphosphatidylinositol (GPI) anchor, and as a result, it has no cytoplasmic domain and no ITSMs [9,10]. SLAMF8/CD353 and SLAMF9/CD84-H1 have only a short cytoplasmic domain and lack ITSMs [11,12].

The majority of SLAMF receptors act as self-ligands that bind via their Ig V-like domains. In contrast, SLAMF2/CD48 binds to SLAMF4/CD244 and vice versa. SLAMF8 and SLAMF9 have no known ligands [1,13,14,15,16,17]. Upon activation of SLAMF receptors, ITSMs can bind SAP, EAT-2, or SH2 domain-containing phosphatases such as protein tyrosine phosphatases SHP-1 and SHP-2, or inositol phosphatase SHIP-1 [18,19]. SAP and the phosphatases compete for the same binding sites [5,20]. The type of cell, the type of SLAMF receptor, and the predominating ITSM-binding ligand will determine which signaling pathways will be activated or inhibited and what will be the resulting biological effect.

Most hematopoietic cells express between 3 and 5 individual SLAMF receptors [21]. The expression of SLAMF receptors by normal B cells has been comprehensively characterized by several research groups [22,23,24,25,26]. Individual SLAMF receptors display different expression patterns during B cell maturation. In general, immature B cells express low levels of SLAMF receptors and their expression gradually increases toward more mature stages, achieving its peak in plasmablasts and plasma cells (e.g., SLAMF1/CD150, SLAMF3/CD229, or SLAMF7/CD319). The expression of SLAMF2/CD48 is highest in peripheral blood B cells and unswitched memory B cells, while the lowest levels are observed in germinal center B cells and plasma cells. The expression of SLAMF5/CD84 is relatively low and stable throughout maturation while SLAMF6/CD352 expression greatly increases after mature B cells leave the bone marrow, and then it remains stably high. SLAMF4/CD244 was found to be completely absent in B cells [24]. 

There is only very limited data on the expression of SLAMF8/CD353 and SLAMF9/CD84-H1 in B cells. The expression of human SLAMF8/CD353 was detected by Northern blot analysis and TaqMan quantitative PCR in dendritic cells and activated monocytes but not in B cells [11]. By flow cytometry, SLAMF8/CD353 was found to be faintly expressed in B cells: only 6-21% of cells within different B cell subsets were weakly positive (in terms of their mean fluorescence intensity, MFI) [25]. Another flow cytometry study reported weak expression of SLAMF8/CD353 in a subset of mature B cells (20–40% positive cells) [27]. SLAMF9/CD84-H1 was found to be present in a fraction of peritoneal CD19+ CD11b+ B1 cells but absent in all peritoneal CD19+ CD11b− B2 cells by flow cytometry. In human PBMCs from healthy donors, a limited SLAMF9/CD84-H1 reactivity in some circulating B cells was observed [28]. As for SAP, its positive expression was detected by RT-PCR in normal B cells [29,30], by flow cytometry in peripheral blood mature B cells [31], in naïve and germinal center B cells in tonsils [32], and by immunohistochemistry in germinal centers and the interfollicular zone of reactive lymph nodes and tonsils [22]. On the other hand, ex vivo and activated B cells were SAP-negative by Western blotting [33]. While the data on SAP expression are not unambiguous, EAT-2 was found to be consistently expressed in B cells [7,31].

The aim of this paper is to provide an overview of the current knowledge on the expression of SLAMF receptors, SAP, and EAT-2 in B-CLPD. SLAMF receptors may be potentially useful as diagnostic or differentially diagnostic markers, prognostic markers, and targets for the development of new drugs for patients with B-CLPD.

## 2. SLAMF1/CD150 (SLAM, IPO-3)

SLAMF1/CD150 is one of the best-studied SLAMF receptors. Among different B-CLPD subtypes, the expression of SLAMF1/CD150 is the most comprehensively studied in CLL.

Yigit et al. (2016) analyzed the expression of most SLAMF receptors in CLL cells isolated from 57 patients. In the majority of cases, the expression of SLAMF1/CD150 was weak and it was present in less than 50% of cells [34].

In a cohort of 300 CLL patients, Bologna et al. (2016) observed that the expression of SLAMF1/CD150 (as analyzed by flow cytometry) was lost in a subset of patients with an aggressive CLL that was associated with a shorter time to first treatment and reduced overall survival. They showed that SLAMF1/CD150 ligation with an agonistic antibody induced autophagy in CLL cells. The molecular cascade connecting SLAMF1/CD150 to the activation of autophagy relies on the accumulation of ROS, the activation of the MAP kinases, and the phosphorylation of BCL2. This last step causes a dissociation between BCL2 and beclin1 and the consequent assembly of the autophagic macrocomplex, which contains SLAMF1 itself, beclin1, and VPS34. SLAMF1/CD150-silenced CLL-like Mec-1 cells or SLAMF1/CD150(lo) primary CLL cells were resistant to autophagy-activating therapeutic agents, such as fludarabine (a part of the FCR chemoimmunotherapy combination used for the treatment of CLL) and ABT-737, a BCL2 inhibitor. The results of Bologna et al. indicate that loss of SLAMF1/CD150 expression in CLL modulates genetic pathways that regulate autophagy and that could potentially negatively affect responses to drugs such as fludarabine or venetoclax, a BCL2 inhibitor recently approved for front-line therapy of CLL. These effects may underlie the unfavorable clinical outcomes experienced by SLAMF1/CD150(lo) patients [35].

Our group also reported the altered expression of SLAMF/CD150 receptors on pathological B cells in peripheral blood samples from 36 patients with CLL [36]. Pathological CLL B cells from patients with CLL showed significantly decreased expression of SLAMF1/CD150 in comparison with normal B cells from healthy control subjects or normal polyclonal B cells present in CLL patient samples [36].

In a series of 349 CLL patients, Rigolin et al. (2021) found that SLAMF1/CD150 downregulation was significantly associated with numerous adverse prognostic factors including highly complex karyotypes, advanced Binet clinical stage, CD38 positivity, high beta2-microglobulin levels, immunoglobulin heavy chain variable region gene (IGHV) unmutated status, 11q deletion, tumor protein p53 (TP53) disruption, and higher risk CLL-International Prognostic Index (CLL-IPI) categories. Downregulated SLAMF1/CD150 levels also had an independent negative prognostic impact on time-to-first treatment and overall survival [37].

The impact of SLAMF1/CD15 and SLAMF7/CD319 on the B cell receptor (BCR) signaling axis was studied by von Wenserski et al. (2021) [38]. They showed significantly decreased degranulation capacity of primary NK cells from CLL patients expressing low levels of SLAMF1/CD150 and SLAMF7/CD319. Overexpression of SLAMF1/CD150 or SLAMF7/CD319 in IGHV mutated CLL cell models resulted in reduced proliferation and impaired responses to BCR ligation, whereas the knockout of both receptors showed opposing effects and increased sensitivity toward inhibition of components of the BCR pathway. Consequently, high levels of SLAMF1/CD150 and SLAMF7/CD319 may lead to lesser therapeutic efficacy of the BCR pathway antagonists such as ibrutinib (or other Bruton kinase inhibitors) in CLL patients with mutated IGHV. The authors also showed that SLAMF1/CD150 and SLAMF7/CD319 receptors mediate their BCR pathway antagonistic effects via recruitment of prohibitin-2 (PHB2), thereby impairing its role in signal transduction downstream of the IGHV-mutant IgM-BCR [38].

Recently, Gordiienko et al. (2024) reported that the sSLAMF1/CD150 isoform was found in all studied plasma samples of CLL patients at different levels regardless of the cell surface SLAMF1/CD150 expression status of B cells and sSLAMF1/CD150 mRNA expression [39]. CLL cases with low levels of the cell surface SLAMF1/CD150 expression in B cells are characterized by high levels of sSLAMF1/CD150 in blood plasma in contrast to CLL cases with high cell surface SLAMF1/CD150 expression in B cells. The elevated levels of sCD150 in blood plasma are associated with a better sensitivity of malignant B cells to cyclophosphamide and bendamustine. The sCD150 isoform is actively secreted by CLL B cells with accumulation in blood plasma, which may be regarded as an additional factor in the CLL clinicopathologic variability [39].

As for other B-CLPD subtypes, Sidorenko et al. (1992), using the IPO-3 antibody, detected reactivity in a number of Burkitt lymphoma (BL) cell lines as well as in 2 out of 30 samples from CLL patients and 4 out of 7 samples from patients with hairy cell leukemia (HCL) [40]. A flow cytometry analysis of Shlapatska et al. (2001) revealed that all studied B lymphoblastoid cell lines (CESS, MP-1, T5-1, 6.16, and RPMI-1788) and cell lines from patients with XLP (IARC 739, XLP-D, XLP-8002, and XLP-8005), expressed high levels of SLAMF1/CD150. The Burkitt lymphoma cell lines Raji, Namalwa 12 (EBV+), and BJAB (EBV−) expressed SLAMF1/CD150 at a moderate level; all other B cell lines tested (REH, Ramos, B104, and RPMI-8226) were SLAMF1/CD150 negative [32]. Another flow cytometry study showed that SLAMF1/CD150 was partially positive in the Namalwa (40–60% cells positive), Raji (60–80%), and Ramos (20–40%) Burkitt lymphoma cell lines and negative only in Daudi (<20%) Burkitt lymphoma B cells [23,24]. Here, it should be noted that the expression of SLAMF1/CD150 in Burkitt lymphoma could depend on the presence of Epstein–Barr virus infection. Indeed, Takeda et al. (2014) observed SLAMF1/CD150 expression in BL-41, BJAB, and Mutu EBV-positive Burkitt lymphoma cells bearing type III latency but not in EBV-negative or type I latency BL41, Akata, BJAB, Daudi, or Mutu Burkitt lymphoma cells [41]. Also, Farage cells (an EBV-positive diffuse large B cell lymphoma (DLBCL) cell line) were found to express high levels of SLAMF1/CD150. High expression of SLAMF1/CD150 in Farage cells correlated with high expression of LMP1, an EBV latent gene. SLAMF1/CD150 (high) cells were found to be responsible for the poor response of Farage cells to CHOP combination chemotherapy [42]. Later, SLAMF1/CD150 was found to contribute to Farage cells’ survival through the Akt signaling pathway [43].

Mikhalap et al. (2004) studied the expression of SLAMF1/CD150 by immunohistochemistry in various types of chronic B cell lymphomas [22]. They found that SLAMF1/CD150 expression was positive in activated B cell DLBCL (ABC-DLBCL) (three out of three cases) but negative in germinal center DLBCL (GC-DLBCL) (nine out of nine). Two out of two cases of mantle cell lymphoma (MCL) displayed low levels of SLAMF1/CD150 expression. On the other hand, all analyzed cases of small lymphocytic lymphoma (SLL) (seven out of seven), lymphoplasmacytic lymphoma (LPL) (two out of two), and sporadic Burkitt lymphoma (three out of three) were SLAMF1/CD150-negative [22].

Fanoni et al. (2011) analyzed the expression of SLAMF1/CD150 by immunohistochemistry in 25 patients with primary cutaneous (PC) B cell lymphomas including 10 PC-follicular center lymphoma (PC-FCL), 10 PC-marginal zone lymphoma (PC-MZL), and 5 PC-DLBCL-LT (leg type) [44]. They found positive expression of SLAMF1/CD150 in PC-FCL while PC-MZL and PC-DLBCL-LT cases were negative.

The expression patterns of all SLAMF receptors, SAP, and EAT-2 are summarized in Table 1.

## 3. SLAMF2/CD48 (BCM1, Blast-1, HuLy-m3)

SLAMF2/CD48 was originally termed B-LAST 1 and found in B cells transformed with EBV virus and in neoplastic B cells in approximately 80% of chronic lymphocytic leukemia cases and 50% of poorly differentiated B cell lymphoma cases. It was also expressed in EBV-positive (e.g., Raji) but not EBV-negative (e.g., Ramos) Burkitt lymphoma cell lines [45]. In the study of de Salort et al. (2011), SLAMF2/CD48 was positive (80–100% positive cells) in Daudi, Namalwa, Raji, and Ramos Burkitt lymphoma B cell lines [24].

Fanoni et al. (2011) observed positive expression of SLAMF2/CD48 in PC-MZL and variable positivity in PC-DLBCL-LT while PC-FCL cases were negative by immunohistochemistry [44].

Wu et al. (2021) analyzed transcriptome data for DLBCL patients extracted from the GSE31312 and GSE10846 datasets in the Gene Expression Omnibus (GEO) database. They identified four immune-related genes (SLAMF2/CD48, IL1RL, PSDM3, and RXFP3) significantly associated with overall survival. They revealed that SLAMF2/CD48 was significantly upregulated in low-risk DLBCL patients whereas IL1RL, PSDM3, and RXFP3 were strongly elevated in high-risk DLBCL patients [46].

In CLL, SLAMF2/CD48 was found to be positive in practically 100% of CLL cells [34]. Coma et al. (2017) observed a downregulated expression of SLAMF2/CD48 in pathological CLL B cells when compared to normal B cells [36]. It is known that SLAMF2/CD48, when expressed by B cells, stimulates cytotoxic activity of NK cells and CD8+ T cells through an interaction with SLAMF4/CD244 [47]. Hence, the downregulation of CD48 may result in reduced NK and CD8+ T cell cytotoxicity and contribute to the compromised immunity observed in CLL patients. Mou et al. (2023) demonstrated that SLAMF2/CD48-expressing feeder cells can promote the proliferation of primary NK cells (which express SLAMF4/CD244) and reduce NK cell apoptosis by activating the p-ERK/BCL2 pathway both in vitro and in vivo without affecting the overall phenotype. Furthermore, NK cells expanded via the engagement of the SLAMF2/CD48-SLAMF4/CD244 axis showed stronger anti-tumor capability and infiltration ability into the tumor microenvironment, leading to an improvement in therapeutic efficiency [48].

In 1984, Greenaway et al. reported the results of a Phase I trial in which they evaluated the effects of WM63 and WM66 antibodies in seven patients with progressive CLL [49]. WM63 is a murine IgM anti-SLAMF2/CD48 antibody that is lytic with human complement. Four CLL patients received increasing daily doses of WM63 intravenously. All patients demonstrated a significant but transient reduction in the number of circulating leukocytes, and no overall effect on disease progression was observed. Three patients receiving large doses of WM63 demonstrated a decline in complement levels during treatment. Two patients developed dose-limiting side effects to WM63. No human anti-mouse antibody responses were documented [49].

Later, Sun et al. (2000) demonstrated that the i.v. injection of both chimeric cHuLym3 and mouse mHu-Lym3 anti-SLAMF2/CD48 antibodies produced significant antitumor responses in the human Raji cell severe combined immunodeficient mouse model [50]. cHuLym3 had more potent activity than mHuLym3 in ADCC assays in vitro, with human PBMCs as effectors. Up to 60% specific cell lysis was observed with cHuLym3 in ADCC assays. It was suggested that anti-SLAMF2/CD48 antibodies may be useful in the treatment of a number of diseases, including lymphoid leukemias and lymphomas [50].

A major concern regarding SLAMF2/CD48 as a therapeutic target is its broad expression by normal lymphocytes and monocytes, which may cause severe cytopenia and immunosuppression when anti-SLAMF2/CD48 mAb is used as a therapeutic drug. The potential hematological toxicity of anti-SLAMF2/CD48 mAb should therefore be very carefully tested at the pre-clinical stage [51].

## 4. SLAMF3/CD229 (Ly-9)

Using flow cytometry, de la Fuente et al. (2001) analyzed the expression of SLAMF3/CD229 in normal human leukocyte populations, hematopoietic cancer cell lines, and several samples from patients with B cell malignancies [52]. The Burkitt lymphoma B cell lines (Daudi and Raji) and the Epstein–Barr virus-transformed B cell lines (CESS and BEN) expressed the highest levels of SLAMF3/CD229, whereas the rest of the B cell lines expressed low levels. Among the BCLPD cases, SLAMF3/CD229 was positive in CLL (12 positive cases out of 15), MCL (5 out of 5), and HCL (1 out of 2) [52].

In the study of Bund et al. (2006), 18 out of 18 CLL patient samples were SLAMF3/CD229-positive by flow cytometry, with a median of 63% positive CLL B cells [53]. What is more, the authors demonstrated that SLAMF3/CD229 is naturally processed and presented as a tumor-associated antigen in primary CLL B cells (unstimulated or CD40L-stimulated). This enabled the expansion of functional IFN gamma-secreting autologous tumor-specific T cells. The authors suggested that SLAMF3/CD229 can be employed for the design of T cell-based immunotherapeutic strategies against CLL and other SLAMF3/CD229-expressing neoplasms [53].

Consistent with the initial observations of de la Fuente et al. (2001), Yigit et al. [34] and our group [36] also observed the positive expression of SLAMF3/CD229 on pathological CLL B cells. Our results showed that SLAMF3/CD229 is significantly upregulated in CLL in comparison with normal B cells [36]. Due to the strong expression of CD229 in CLL cells, anti-CD229 strategies may be proposed as potential treatments for CLL.

The prognostic impact of SLAMF3/CD229 in CLL was evaluated in a proteomic study by Saberi Hosnijeh et al. (2020) [54]. They analyzed the potential prognostic impact of 360 proteomic markers, including SLAMF3/CD229, on the clinical outcomes of 51 elderly patients and young frail patients with advanced CLL treated with front-line chemoimmunotherapy involving chlorambucil, rituximab, and lenalidomide. The proteomic analysis of pre-treatment serum samples revealed that patients with high levels of SLAMF3/CD229 (higher than the median) had a shorter event-free survival (EFS) than those with SLAMF3/CD229 levels below the median. SLAMF3/CD229 was identified as a promising independent prognostic proteomic marker in patients treated for CLL. However, the authors observed no significant effect of established prognostic factors (beta2-microglobulin, Rai stage, or chromosomal aberrations) on EFS, which was most likely due to the relatively small number of patients in this study [54].

SLAMF3/CD229 expression in BL cell lines was also analyzed by Romero et al. (2004) who found that SLAMF3/CD229 was positive in Daudi (75–100% positive cells) and partially positive in Namalwa (20–50%), Raji (50–75%), and Ramos (20–50%) cells [23]. Similar results were also reported by de Salort et al. (2011) who observed that SLAMF3/CD229 was positive in Daudi (80–100% positive cells) and partially positive in Namalwa (40–60%), Raji (60–80%), and Ramos (20–40%) cells [24].

In primary cutaneous B cell lymphomas, Fanoni et al. (2011) observed positive expression of SLAMF3/CD229 in all analyzed types of primary cutaneous B cell lymphomas, namely PC-FCL, PC-MZL, and PC-DLBCL-LT, by immunohistochemistry [44].

Using tumor tissue microarrays, Roncador et al. (2022) analyzed the expression of SLAMF3/CD229 in 205 patients with nine different subtypes of B-CLPD. SLAMF3/CD229 was expressed in 20% of BL cases, 52% non-GC-DLBCL, 60% FL, 66% GC-DLBCL, 70% CLL, 70% MCL, 82% nodal MZL, 86% mucosa-associated lymphoid tissue (MALT) lymphoma, and 87% splenic MZL. The expression of SLAMF3/CD229 was particularly strong in lymphomas of marginal zone origin [55].

Recently, Li et al. (2022) reported that SLAMF3 and SLAMF4 are immune checkpoints that constrain macrophage phagocytosis of hematopoietic tumors [56]. They found that SLAMF receptor deficiency triggered macrophage phagocytosis of hematopoietic cells. SLAMF3/CD229 and SLAMF4/CD244 were identified as “don’t eat me” receptors on macrophages. These receptors inhibited “eat me” signals. Their loss potently elicited macrophage rejection of hematopoietic tumors. Deletion of SLAMF receptors also significantly enhanced the phagocytosis of CD19-positive hematopoietic targets by macrophages expressing the chimeric CD19 antigen receptor. Therefore, SLAMF receptor-mediated inhibition of macrophage phagocytosis is critical to hematopoietic homeostasis, and SLAMF receptors may represent previously unknown targets for tumor immunotherapy [56].

## 5. SLAMF4/CD244 (2B4, NAIL)

SLAMF4/CD244 is not expressed by normal B cells. In an EBV-transformed B cell line, CESS, Romero et al. (2004) observed 20–50% SLAMF4/CD244-positive cells by flow cytometry. All analyzed Burkitt lymphoma cell lines (Raji, Daudi, Ramos, and Namalwa) were SLAMF4/CD244-negative [23]. Another flow cytometry study confirmed the absence of SLAMF4/CD244 in Daudi, Namalwa, Raji, and Ramos Burkitt lymphoma B cell lines [24].

According to our results, SLAMF4/CD244 was negative by flow cytometry in peripheral blood B cells from healthy control subjects, normal polyclonal B cells found in peripheral blood samples from CLL patients, and pathological CLL B cells from patients with CLL [36].

## 6. SLAMF5/CD84 (Ly9b, GR6)

Several groups analyzed the expression of SLAMF5/CD84 in CLL [34,36,57]. In the study of Binsky-Ehrenreich et al. (2014), low levels of SLAMF5/CD84 mRNA were detected in normal B cells, whereas elevated levels of SLAMF5/CD84 mRNA were observed in all of the CLL patients, regardless of the stage of disease. By flow cytometry, SLAMF5/CD84 cell surface levels were significantly higher in all CLL cells when compared with total or CD5+ normal B cells. Activation of cell surface SLAMF5/CD84 initiated a signaling cascade (involving EAT-2, Lck, Akt, Bcl-2, and Mcl-1) that enhanced CLL cell survival. Both downmodulation of CD84 expression and its immune-mediated blockade induced cell death in vitro and in vivo. The expression of SLAMF5/CD84 was found to be regulated by macrophage migration inhibitory factor (MIF) and its receptor, CD74. CLL patients from an ongoing clinical trial who were treated with humanized anti-CD74 (milatuzumab) showed a decrease in SLAMF5/CD84 mRNA and protein levels in milatuzumab-treated cells. This downregulation was correlated with a reduction in Bcl-2 and Mcl-1 expression [57]. These data show that overexpression of CD84 in CLL is an important survival mechanism that appears to be an early event in the pathogenesis of the disease.

Marom et al. (2016) studied the role of SLAMF5/CD84 in the tumor microenvironment. SLAMF5/CD84 receptors expressed by CLL cells interact with SLAMF5/CD84 expressed by cells in their microenvironment, inducing cell survival on both sides. Blocking SLAMF5/CD84 in vitro and in vivo disrupts the interaction of CLL cells with their microenvironment, resulting in induced cell death [58].

Later, Lewinsky et al. (2018) showed that cell–cell interactions mediated through human and mouse SLAMF5/CD84 upregulated PD-L1 expression in CLL cells and in their microenvironment and PD-1 expression in T cells. This resulted in the suppression of T cell responses and activity in vitro and in vivo. These results demonstrate a role for SLAMF5/CD84 in the regulation of immune checkpoints by CLL cells [59].

The upregulated expression of SLAMF5/CD84 by CLL B cells in comparison with normal B cells was also confirmed by the flow cytometric analysis of Coma et al. [36]. The above findings suggest novel therapeutic strategies for CLL based on the blockade of SLAMF5/CD84 receptors.

In Burkitt lymphoma cell lines, Romero et al. (2004) observed 75-100% SLAMF5/CD84-positivity in Raji, Daudi, Ramos, and Namalwa cells by flow cytometry. Similar results were reported by de Salort et al. (2011): SLAMF5/CD84 was positive (80–100% positive cells) in Namalwa, Raji, and Ramos, but only partially positive (40–60%) in Daudi Burkitt lymphoma cells [24].

Using single-cell RNA sequencing (scRNA-seq) and mass cytometry (CyTOF), Shi et al. (2022) identified a diagnostic biomarker panel comprising 12 biomarkers, including SLAMF5/CD84, that were overexpressed in 18 patients with relapsed/refractory DLBCL versus 5 healthy volunteers [60].

## 7. SLAMF6/CD352 (NTB-A, Ly108, SF2000, TCOM)

Korver et al. (2007) reported that SLAMF6/CD352 expression levels were found to be higher in B cells obtained from both CLL patients and normal volunteers compared with NK and T cells (as assessed by flow cytometry). No statistically significant difference was observed between normal B cells and CLL B cells [61]. In contrast, our group reported an increased expression of SLAMF6/CD352 in CLL B cells in comparison with normal B cells from healthy control subjects. The expression of SLAMF6/CD352 by CLL B cells was also upregulated in comparison with normal polyclonal B cells present in the samples from CLL patients [36]. Korver et al. also demonstrated that anti-SLAMF6/CD352 antibodies induce complement-dependent cytotoxicity (CDC) primarily in B cells isolated from CLL patients and B lymphoma cell lines. Furthermore, anti-SLAMF6/CD352 monoclonal antibodies demonstrated anti-tumor activity against CA46 human lymphoma xenografts in nude mice and against systemically disseminated Raji human lymphoma cells in severe combined immunodeficient mice [61]. These results validate SLAMF6/CD352 as a potential target for immunotherapy of B cell leukemias and lymphomas.

Later, Yigit et al. (2016) reported that upon transplantation of an aggressive murine B220+ CD5+ CLL cell clone, TCL1-192, into SCID mice, one injection of an anti-SLAMF6/CD352 monoclonal antibody abrogated tumor progression in the spleen, bone marrow, and blood, but not in the peritoneal cavity or omentum [34]. Similarly, the progression of a murine B cell lymphoma, LMP2A/λMyc, was also eliminated by an anti-SLAMF6/CD352 monoclonal antibody. Co-administering anti-SLAMF6/CD352 monoclonal antibody with the Bruton tyrosine kinase inhibitor, ibrutinib, synergized to efficiently eliminate the tumor cells in the spleen, bone marrow, liver, and the peritoneal cavity. Moreover, an anti-human SLAMF6/CD352 monoclonal antibody efficiently killed human CLL cells both in vitro (in combination with ibrutinib, it induced apoptosis in OSU-CLL cells) and in vivo (it reduced the growth of MEC-1 cells transplanted subcutaneously into [Rag x γc]^−/−^ mice). The authors proposed that a combination of anti-SLAMF monoclonal antibody with ibrutinib should be considered a novel therapeutic approach for CLL and other B cell tumors [34].

Yigit et al. (2019) also assessed the role of SLAMF6/CD352 as an immune checkpoint regulator potentially capable of overcoming T-cell exhaustion in CLL [62]. Using mouse models, they observed that the transfer of SLAMF6/CD352+ Eμ-TCL1 CLL cells into SLAMF6/CD352^−/−^ recipients, in contrast to wild-type (WT) recipients, significantly induced the expansion of a PD-1+ subpopulation among CD3+CD44+CD8+ T cells, which had impaired cytotoxic functions. Conversely, administering anti-SLAMF6/CD352 significantly reduced the leukemic burden in Eμ-TCL1 recipient WT mice concomitantly with a loss of PD-1+CD3+CD44+CD8+ T cells with significantly increased effector functions. Anti-SLAMF6/CD352 significantly reduced the leukemic burden in the peritoneal cavity, a niche where ADCC is impaired, possibly through activation of CD8+ T cells. In vitro exhausted CD8+ T cells showed increased degranulation when anti-human SLAMF6/CD352 was added to the culture. Taken together, anti-SLAMF6/CD352 both effectively corrected CD8+ T-cell dysfunction and had a direct effect on tumor progression. Targeting SLAMF6/CD352 can be suggested as a potential therapeutic strategy for CLL [62].

Korver et al. (2007) also analyzed the expression of SLAMF6/CD352 in various cell lines by Western blotting and found the highest level of SLAMF6/CD352 expression in B cell lines (CA46, Daudi, Raji, and Ramos). Lower levels were detected in Jurkat T cells, while SLAMF6/CD352 was not detected in myeloid-derived cells (THP-1, U939, and K562). By immunohistochemistry, SLAMF6/CD352 was detected in NHL samples: positive staining was observed in DLBCL, SLL, MCL, and FL samples [61]. The expression of SLAMF6/CD352 by Burkitt lymphoma cell lines was also confirmed by flow cytometry with SLAMF6/CD352 being positive on 80-100% of Daudi, Namalwa, Raji, and Ramos Burkitt lymphoma cells [24].

## 8. SLAMF7/CD319 (CS1, CRACC, 19A24)

Using RT-PCR, Lee et al. (2007) found that human B cells only expressed the long isoform of SLAMF7/CD319, CS1-L, which contains ITSMs in its cytoplasmic domain (the short isoform does not contain ITSMs). Its expression was upregulated upon B cell activation with various stimulators. Anti-SLAMF7/CD319 monoclonal antibody strongly enhanced the proliferation of both freshly isolated and activated B cells, especially when B cells were activated by anti-CD40 monoclonal antibodies and/or human recombinant IL-4. The effects of SLAMF7/CD319 on B cell proliferation were shown on both naive and memory B cells. SLAMF7/CD319 activation enhanced mRNA transcripts of flt3 ligand, lymphotoxin A, TNF, and IL-14. Neutralizing antibodies against lymphotoxin A, TNF-alpha, and/or flt3 ligand abolished the effect of SLAMF7/CD319 on B cell proliferation. These results suggest that activation of B cells, through surface SLAMF7/CD319, may be mediated through the secretion of autocrine cytokines and SLAM7/CD319 may play a role in the regulation of B lymphocyte proliferation during immune responses [63].

Among B-CLPD, the expression of SLAMF7/CD319 was observed in CLL, albeit weak [34,36]. By flow cytometry analysis, pathological CLL B cells showed a decreased expression of SLAMF7/CD319 in comparison with normal B cells from healthy control subjects as well as when compared with normal polyclonal B cells present in CLL patient samples [36].

SLAMF7/CD319, along with SLAMF1/CD150, was shown to affect the degranulation capacity of primary NK cells from CLL patients and have an antagonistic effect on the BCR pathway via recruitment of prohibitin-2 (PHB2) [38].

SLAMF7/CD319 was also found to be positive in Raji (80–100% positive cells) and partially positive in Daudi (40–60%), Namalwa (60–80%), and Ramos (60–80%) Burkitt lymphoma B cell lines [24].

The expression of SLAMF7/CD319 was also reported in plasmablastic lymphoma (PBL) [64]. Shi et al. (2018) used immunohistochemistry to analyze 20 PBL cases, including 11 PBL not otherwise specified (PBL NOS), 5 HIV+ PBL, 2 ALK+ large B cell lymphomas (LBCL), 1 primary effusion lymphoma, and 1 PBL with angioimmunoblastic T cell lymphoma (AITL). Overall, 85% cases were found to be positive for SLAMF7/CD319. Among PBL NOS and ALK+ LBCL, 82% and 50% of cases were SLAMF7/CD319-positive, respectively. All remaining PBL subtypes were also SLAMF/CD319-positive. The authors have also detected the expression of SLAMF7/CD319 in BC2, a PBL/PEL cell line, by flow cytometry and immunohistochemistry. The intensity of SLAMF7/CD319 expression in BC2 cells was comparable to that of OPM2 multiple myeloma cells. Elotuzumab, a therapeutic anti-SLAMF7/CD319 antibody, induced dose-dependent ADCC activity toward BC2 target cells, using PBMCs as effector cells. Elotuzumab also enhanced granzyme B release from PBMC effector cells against BC2 cells [64].

Panaampon et al. (2022) analyzed the expression of SLAMF7/CD319 by flow cytometry in a panel of primary effusion lymphoma cell lines (BCBL-1, BC-1, BC-2, BC-3, RM-P1, GTO, and TY-1), multiple myeloma cell lines (KMM-1, KMS-12-PE, RPMI-8226, U266, MM1.S, MM1.R, KMS-11, NCI-H929, JJN-3, and IM-9) and a mantle cell lymphoma (MCL) cell line (Z138) [65]. All seven PEL lines expressed consistently high levels of SLAMF7/CD319. Compared to PEL cell lines, the expression of SLAMF7/CD319 was weaker in MM cell lines and normal B cells from a healthy donor, while Z138 mantle lymphoma cells were found to be SLAMF7/CD319-negative. Furthermore, the authors showed that elotuzumab demonstrated potent ADCC against PEL using expanded NK cells as effector cells. In the process of ADCC, NK cells increased the surface expression of CD107a. Elotuzumab also enhanced the survival of PEL-bearing immunodeficient mice with the adoptive transfer of human NK cells [65].

Thus, SLAMF7/CD319 may be a useful marker for the diagnosis and characterization of PBL as well as a potential drug target in PBL. Elotuzumab could potentially be used for the treatment of SLAMF/CD319-positive chronic B cell lymphoproliferative disorders, including CLL and PBL/PEL.

## 9. SLAMF8/CD353 (BLAME, SBBI42)

Currently, very little data are available on the expression of SLAMF8/CD353 in B-CLPD. In two analyzed CLL cell lines, SLAMF8/CD353 was negative in MEC-1 cells, while its expression in OSU-CLL cells could be regarded as very weakly positive [34].

## 10. SLAMF9/CD84-H1 (SF2001, CD2F10)

The receptor SLAMF9/CD84-H1 was discovered by Zhang et al. (2001) and its expression (by Northern blot analysis) was restricted to hematopoietic tissues (spleen, lymph nodes, peripheral blood, and bone marrow). RT-PCR analysis showed the expression of SLAMF9/CD84-H1 in dendritic cells, monocytes, the THP-1 monocytic cell line, Jurkat, and HuT78 T-lymphoid cell lines as well as Raji and Ramos Burkitt lymphoma cell lines [66].

## 11. SAP–SLAM-Associated Protein (SH2D1A)

By Western blot analysis, all three EBV-positive type I Burkitt lymphoma cell lines (Rael, Akata and Mutu I, clones 148 and 216) were SAP positive. Four type III BL lines (Raji, Daudi, P3HR1, and Namalwa), seven EBV-negative BL lines (BL41, BL49, BL28, DG75, BL2, CA 46, and Ramos) and two EBV-negative B-cell lymphoma lines (JD38 and Bjab) were all SAP negative. The phenotypically type III convertants (BL41/95, BL28/Ak) of originally EBV-negative BL lines were SAP negative as well. The Mutu III subline expressed SAP at a low level [33].

Another Western blot analysis of whole cell lysates of 15 different B lymphoblastoid and B lymphoma cell lines revealed that SAP was expressed in two out of the five B lymphoblastoid cell lines studied, MP-1 and CESS. Flow cytometry also showed intracellular expression of SAP in MP-1, but not in the BJAB Burkitt lymphoma cell line. Immunohistochemical analysis revealed SAP expression in frozen sections from tonsils, lymph nodes, and DLBCL. Furthermore, SAP protein levels in MP-1 cells were upregulated by CD40 crosslinking, downregulated by B cell receptor ligation, and unaffected by CD150 crosslinking [32].

SAP (DSHP) expression in DLBCL was also observed by RNA in situ hybridization [30].

By immunohistochemistry, SAP was positive in 12 out of 12 analyzed DLBCL cases, including 9 GC-DLBCL and 3 ABC-DLBCL, as well as in 14 out of 14 cases of Hodgkin lymphoma. In contrast, all analyzed cases of SLL (seven patients), sporadic Burkitt lymphoma (three cases), MCL (two cases), and LPL (two cases) were SAP-negative [22].

In another immunohistochemistry study, SAP tested negative in all cases of lymphoblastic B-NHL (0 positive cases out of 11), CLL (0/20), MCL (0/20), FL grade 1, 2, and 3 (0/114), Burkitt lymphoma (0/14), nodal and splenic MZL (0/23), MALT (0/12), and HCL (0/1). In DLBCL, a sporadic expression of SAP (3/115) was observed [67].

In CLL, SAP expression could not be detected at the mRNA level [57].

## 12. EAT-2—Ewing’s Sarcoma Transcript-2 (EWS/FLI1-Activated Transcript 2, SH2D1B)

EAT-2 was found to be expressed in antigen-presenting cells such as B lymphocytes and macrophages. Human EAT-2 nucleotide sequences were amplified using RNA from five out of the six B lymphoma or lymphoblastoid cell lines tested (CESS, Daudi, Namalwa, Raji, and RPMI-1888) [7].

In another study, EAT-2 mRNA and protein were expressed in normal peripheral B cells as well as in CLL cells [57].

## 13. Conclusions

In the present paper, we have summarized the current knowledge on the expression patterns of SLAMF receptors and their adaptor proteins SAP and EAT-2 in various types of B-CLPD (Table 1). Among the SLAMF receptors, the expression of SLAMF1/CD150 can be considered the best described in B-CLPD. Its expression was detected in CLL (but not SLL), ABC-DLBCL (but not GC-DLBCL), PC-FCL, MCL, HCL, and BL, while PC-MZL and LPL were negative. SLAMF2/CD48 was positive in CLL, DLBCL, PC-MZL, and BL, while it was negative in PC-FCL. The expression of SLAMF3/CD229 was observed in CLL, DLBCL, FL, MCL, MZL, HCL, and BL. SLAMF5/CD84 was detected in CLL, DLBCL, and BL. SLAMF6/CD352 was expressed in a number of B-CLPD types including CLL, SLL, DLBCL, FL, MCL, and BL. SLAMF7/CD319 expression was found in CLL, PBL/PEL, and BL, while MCL was negative. SLAMF4/CD244 was not expressed by B cells and it was found to be negative in CLL and BL. There are currently no data on the expression of SLAMF8/CD353 in B-CLPD and SLAMF9/CD84-H1 has been analyzed only in BL. The adaptor protein SAP was detected in DLBCL and BL, but CLL, SLL, FL, MCL, MZL, LPL, and HCL were negative. EAT-2 was found to be positive in CLL and BL, and the remaining types of B-CLPD have not been tested for EAT-2.

Among the B-CLPD types, the expression patterns of SLAMF receptors are the most comprehensively characterized in CLL. Pathological CLL B cells showed weak expression of SLAMF1/CD150 and SLAMF7/CD319. The receptors SLAMF2/CD48, SLAMF3/CD229, SLAMF5/CD84, and SLAMF6/CD352 were expressed with medium to strong intensity. SLAMF1/CD150, SLAMF2/CD48, and SLAMF7/CD319 were found to be downregulated while SLAMF3/CD229, SLAMF5/CD84, and SLAMF6/CD352 were overexpressed in CLL. SLAMF5/CD84 and SLAMF7/CD319 were also upregulated in DLBCL and PBL/PEL, respectively.

An obvious limitation of the above conclusions is that the data were derived from studies that used different methods to assess the expression of SLAMF receptors. Also, the analyzed biological materials included not only B-CLPD patient samples but also B-lymphoid cancer cell lines (especially Burkitt lymphoma cell lines). Nevertheless, the observed expression patterns of SLAMF receptors might be potentially exploitable for the diagnosis or differential diagnosis of B-CLPD (e.g., strong expression of SLAMF7/CD319 was found to be typical of PBL/PEL, while MZL was characterized by its strong expression of SLAMF3/CD229). Some SLAMF receptors have been demonstrated to possess prognostic significance. For instance, in CLL, low expression of SLAMF1/CD150 by pathological B cells was associated with an adverse prognosis, whereas in DLBCL, its high expression was linked to resistance to chemotherapy. Also, CLL patients with high expression of SLAMF3/CD229 have a significantly shorter EFS. Several SLAMF receptors including SLAMF2/CD48, SLAMF5/CD84, SLAMF6/CD352, and SLAMF7/CD319 have become interesting druggable targets. Potential anti-tumor effects of anti-SLAMF antibodies against B-CLPD have been tested both in vitro and in vivo.

To conclude, more research is necessary to fully characterize the expression of SLAMF receptors, SAP, and EAT-2 in B-CLPD. In many types of B-CLPD, their expression patterns have not been analyzed yet. Nevertheless, several SLAMF receptors could be regarded as potential diagnostic and differential diagnostic markers, prognostic factors, and targets for the development of novel drugs for patients with B-CLPD.

## Figures and Tables

**Figure 1 ijms-25-04014-f001:**
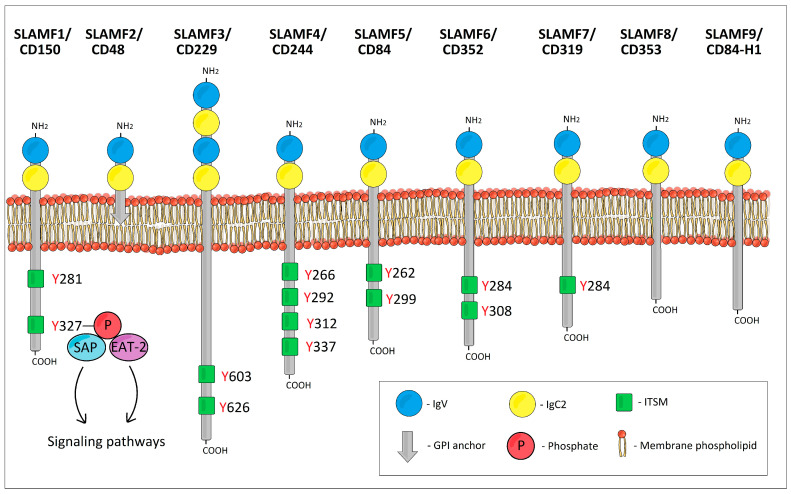
The schematic structure of SLAM family receptors.

**Table 1 ijms-25-04014-t001:** Summary of the expression of SLAMF receptors, SAP, and EAT-2 in chronic lymphoproliferative disorders (based on data from flow cytometry, immunohistochemistry, Western blotting, and mRNA expression studies in cancer cell lines and samples from patients with B-CLPD).

B-CLPD Type:	CLL	SLL	DLBCL	FL	MCL	MZL	LPL	HCL	PBL/ PEL	BL
SLAMF1/CD150	+ down	−	+ (ABC) − (GC) − (PC, LT)	+ (PC)	+	− (PC)	−	+	nr	− or +
SLAMF2/CD48	+ down	nr	+ + var (PC)	− (PC)	nr	+ (PC)	nr	nr	nr	+
SLAMF3/CD229	+ up	nr	+	+	+	+	nr	− or +	nr	+
SLAMF4/CD244	−	nr	nr	nr	nr	nr	nr	nr	nr	−
SLAMF5/CD84	+ up	nr	+ up	nr	nr	nr	nr	nr	nr	+
SLAMF6/CD352	+ up	+	+	+	+	nr	nr	nr	nr	+
SLAMF7/CD319	+ down	nr	nr	nr	-	nr	nr	nr	+ up	+
SLAMF8/CD353	+?	nr	nr	nr	nr	nr	nr	nr	nr	nr
SLAMF9/CD84-H1	nr	nr	nr	nr	nr	nr	nr	nr	nr	+
SAP	−	−	+ (ABC) + (GC)	−	−	−	−	−	nr	− or +
EAT-2	+	nr	nr	nr	nr	nr	nr	nr	nr	+

B-CLPD types: CLL, chronic lymphocytic leukemia; SLL, small lymphocytic lymphoma; DLBCL, diffuse large B cell lymphoma; FL, follicular lymphoma; MCL, mantle cell lymphoma; MZL, marginal zone lymphoma; LPL, lymphoplasmacytic lymphoma; HCL, hairy cell leukemia; PBL, plasmablastic lymphoma; PEL, primary effusion lymphoma; BL, Burkitt lymphoma. Subtypes: ABC, activated B cell; GC, germinal center; PC, primary cutaneous; LT, leg type. Antigen expression: + positive; − negative; up, upregulated; down, downregulated; var, variable; nr, not reported.

## Data Availability

Not applicable.

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
