# Peer review of "SLAM Family Receptors in B Cell Chronic Lymphoproliferative Disorders"

_ijms, 2024, doi:10.3390/ijms25074014_

Round 1

Reviewer 1 Report

Comments and Suggestions for Authors

The review by Dominik Kľoc and collaborators provides a comprehensive overview of the immunoreceptor SLAM family receptors. The manuscript synthesizes published findings regarding the expression of these molecules on B cell malignancies and its potential as prognostic markers and therapeutic targets. Overall, the manuscript is well-written.

However, a weakness of this review is the authors' limited expertise in the study of these molecules, as evidenced by their single publication on this family of receptors (reference number 34). This limitation impacts the discussion of conflicting results found in the literature.

A major point for improvement is the need for the authors to elaborate on the consequences of the co-expression of different SLAMF receptors and adaptor proteins (SAP and EAT2). This would aid in better interpreting the functional significance of these receptors for tumor B cell proliferation and survival. Without this, the risk is that the paper may overly focus on these molecules as biomarkers, neglecting their potential functional roles in tumorigenesis and subsequent modulation for therapy.

Some suggestions for improvement include:

-The authors should revise this paper about the prognostic value of CD150:

Saberi Hosnijeh F, van der Straten L, Kater AP, van Oers MHJ, Posthuma WFM, Chamuleau MED, Bellido M, Doorduijn JK, van Gelder M, Hoogendoorn M, de Boer F, Te Raa GD, Kerst JM, Marijt EWA, Raymakers RAP, Koene HR, Schaafsma MR, Dobber JA, Tonino SH, Kersting SS, Langerak AW, Levin MD. Proteomic markers with prognostic impact on outcome of chronic lymphocytic leukemia patients under chemo-immunotherapy: results from the HOVON 109 study. Exp Hematol. 2020 Sep;89:55-60.e6. doi: 10.1016/j.exphem.2020.08.002.

-The authors could revise and add the following references related to Ly9 about this molecule as a tumor-associated antigen in CLL and its expression of on different B cell leukemias and lymphomas.

Bund D, Mayr C, Kofler DM, Hallek M, Wendtner CM. Human Ly9 (CD229) as novel tumor-associated antigen (TAA) in chronic lymphocytic leukemia (B-CLL) recognized by autologous CD8+ T cells. Exp Hematol. 2006 Jul;34(7):860-9. 

Roncador G, Puñet-Ortiz J, Maestre L, Rodríguez-Lobato LG, Jiménez S, Reyes-García AI, García-González Á, García JF, Piris MÁ, Montes-Moreno S, Rodríguez-Justo M, Mena MP, Fernández de Larrea C, Engel P. CD229 (Ly9) a Novel Biomarker for B-Cell Malignancies and Multiple Myeloma. Cancers (Basel). 2022 Apr 26;14(9):2154. 

-The authors are suggested to check the supplementary material of reference #58 that contains detailed expression results of SLAMF molecules on B-CLL cells from a large set of patients.

-Table 1 is not very informative since most of the information is missing for most of the SLAMF molecules. The authors could eliminate it.

Reviewer 2 Report

Comments and Suggestions for Authors

1. Have studies explored the functional implications of SLAMF receptor expression in cancer cell lines derived from B-lymphoid origin, particularly regarding their involvement in tumorigenesis or disease progression?

2. Are there any ongoing investigations or clinical trials exploring novel drugs targeting SLAMF receptors in B-CLPD, and how might these therapeutic interventions potentially impact patient outcomes?

3. What are the functional differences between typical SLAMF receptors and atypical ones like SLAMF2/CD48, SLAMF8/CD353, and SLAMF9/CD84-H1 in hematopoietic cell biology, and how do these differences influence their roles?

4. While most SLAMF receptors act as self-ligands, are there known ligands for atypical receptors like SLAMF8 and SLAMF9, and what are the implications for their biological functions?

5. Are there any ongoing or emerging studies shedding light on the roles of SLAMF8/CD353 and SLAMF9/CD84-H1 in B cell biology, given the limited data discussed in the review?

6. How do the expression patterns of SLAMF receptors vary among different subtypes of B-CLPD, and how might these variations impact disease classification and patient management in terms of diagnosis and prognosis?

7. Can the review provide comprehensive insights into the significance of SLAMF1/CD150 expression across various B-CLPD subtypes, emphasizing its association with disease aggressiveness and clinical outcomes?

8. Considering the association between SLAMF1/CD150 expression levels and clinical prognostic factors in CLL, are there potential implications for tailoring risk stratification and personalized treatment approaches in CLL patients?

9. Could the review delve deeper into the molecular mechanisms underlying the effects of altered SLAMF1/CD150 expression on genetic pathways regulating chemotaxis and autophagy, and how might this knowledge inform therapeutic strategies?

10. Given the impact of SLAMF1/CD150 downregulation on the BCR signaling axis, how might this influence the efficacy of targeted therapies or immunotherapies that target BCR signaling pathways in B-CLPD?

11. Regarding soluble SLAMF1/CD150 levels in CLL clinicopathologic variability, can the review elaborate on the mechanisms underlying its secretion by CLL B cells and its clinical implications?

12. Are there known functional implications of SLAMF1/CD150 expression in other B-CLPD subtypes, such as Burkitt lymphoma, mantle cell lymphoma, and primary cutaneous B cell lymphomas, and how do these findings contribute to our understanding of disease pathogenesis and progression?

13. How do the findings on SLAMF1/CD150 expression in different B-CLPD subtypes contribute to our broader understanding of the roles of SLAMF receptors in hematological malignancies, and what are the implications for future research or clinical practice?

14. Overall, how effectively does the review synthesize the current knowledge on SLAMF1/CD150 expression in various B-CLPD subtypes, and what are the key implications for clinicians and researchers in the field?

Comments on the Quality of English Language

Thank you for the opportunity to review your manuscript. Overall, the English language in your work is commendable
